# In Situ Cosmogenic $^{10}$Be Dating of Laurentide Ice Sheet Retreat from Central New England, USA

**Jason S. Drebber** [1] , **Christopher T. Halsted** [1] , **Lee B. Corbett** [1] , **Paul R. Bierman** [1,*] and **Marc W. Caffee** [2]

1   Rubenstein School of Environment and Natural Resources, University of Vermont,
    Burlington, VT 05403, USA; jason.drebber@uvm.edu (J.S.D.); chalsted@uvm.edu (C.T.H.);
    ashley.corbett@uvm.edu (L.B.C.)

2   Department of Physics and Astronomy and Department of Earth, Atmospheric, and Planetary Sciences,
    Purdue University, West Lafayette, IN 46202, USA; mcaffee@purdue.edu

*   Correspondence: paul.bierman@uvm.edu

**Abstract:** Constraining the timing and rate of Laurentide Ice Sheet (LIS) retreat through the northeastern United States is important for understanding the co-evolution of complex climatic and glaciologic events that characterized the end of the Pleistocene epoch. However, no in situ cosmogenic $^{10}$Be exposure age estimates for LIS retreat exist through large parts of Connecticut or Massachusetts. Due to the large disagreement between radiocarbon and $^{10}$Be ages constraining LIS retreat at the maximum southern margin and the paucity of data in central New England, the timing of LIS retreat through this region is uncertain. Here, we date LIS retreat through south-central New England using 14 new in situ cosmogenic $^{10}$Be exposure ages measured in samples collected from bedrock and boulders. Our results suggest ice retreated entirely from Connecticut by 18.3 ± 0.3 ka (n = 3). In Massachusetts, exposure ages from similar latitudes suggest ice may have occupied the Hudson River Valley up to 2 kyr longer (15.2 ± 0.3 ka, average, n = 2) than the Connecticut River Valley (17.4 ± 1.0 ka, average, n = 5). We use these new ages to provide insight about LIS retreat timing during the early deglacial period and to explore the mismatch between radiocarbon and cosmogenic deglacial age chronologies in this region.

**Keywords:** cosmogenic nuclides; Laurentide Ice Sheet; deglacial chronology; geochronology; beryllium-10; New England

## 1. Introduction

Accurately constraining Laurentide Ice Sheet (LIS) retreat is important for understanding the co-evolution of the inter-related climatic, oceanographic, and glacial events during the late glacial period [1], the impact of LIS deglaciation on ocean water volume and thus sea level [2,3], and local impacts on landscape evolution [4]. The southeastern LIS covered the New England region of the northeastern United States during Marine Isotope Stage 2 ([5]; Figure 1), expanding to its southernmost extent by at least 27 to 24 ka (thousand years ago) [6–10]. Following the Last Glacial Maximum (LGM), the LIS margin retreated northward through New England [5,11]. The exact timing of the LIS retreat initiation in New England is debated but is often interpreted to have started by at least 20 ka [1,5,12], despite persistent stadial conditions in the Northern Hemisphere until approximately 17 ka [13]. Complicating the uncertainty is a 10 kyr difference between the organic $^{14}$C ages and in situ $^{10}$Be exposure ages estimating LIS retreat initiation near its terminal moraine [10,14,15].

Despite the relatively large number of glacier chronology studies in New England, a lack of $^{10}$Be exposure dating through most of Connecticut and Massachusetts makes LIS retreat timing (and corresponding timing to regional climate) less certain [10,16]. Glacial retreat in central New England is dated only by organic $^{14}$C in central Massachusetts [17–19] and the New North American Varve Chronology (NAVC) in the Connecticut River Valley

(Figure 2; [12]). There are no [10]Be age constraints between the Ledyard Moraine (abandoned 21.4 ± 0.7 ka; Figure 1) and the Old Saybrook Moraine (abandoned 21.3 ± 0.9 ka; Figure 1) in southern Connecticut [1] and Mt. Greylock (15.2 ± 1.4 ka; Figure 1) in northern Massachusetts [16]. This leaves a spatial gap in the [10]Be chronology of approximately 125 km and more than 6 kyr, coinciding with the timing of LIS retreat during a changing late glacial New England climate [20,21]. Furthermore, the presence of two different ice lobes (one in the Hudson and the other in Connecticut River Valley, separated by the Berkshire Massif in Massachusetts) likely leads to more complexity in LIS marginal positions than is represented by larger-scale reconstructions (e.g., [5,11]; Figure 1).

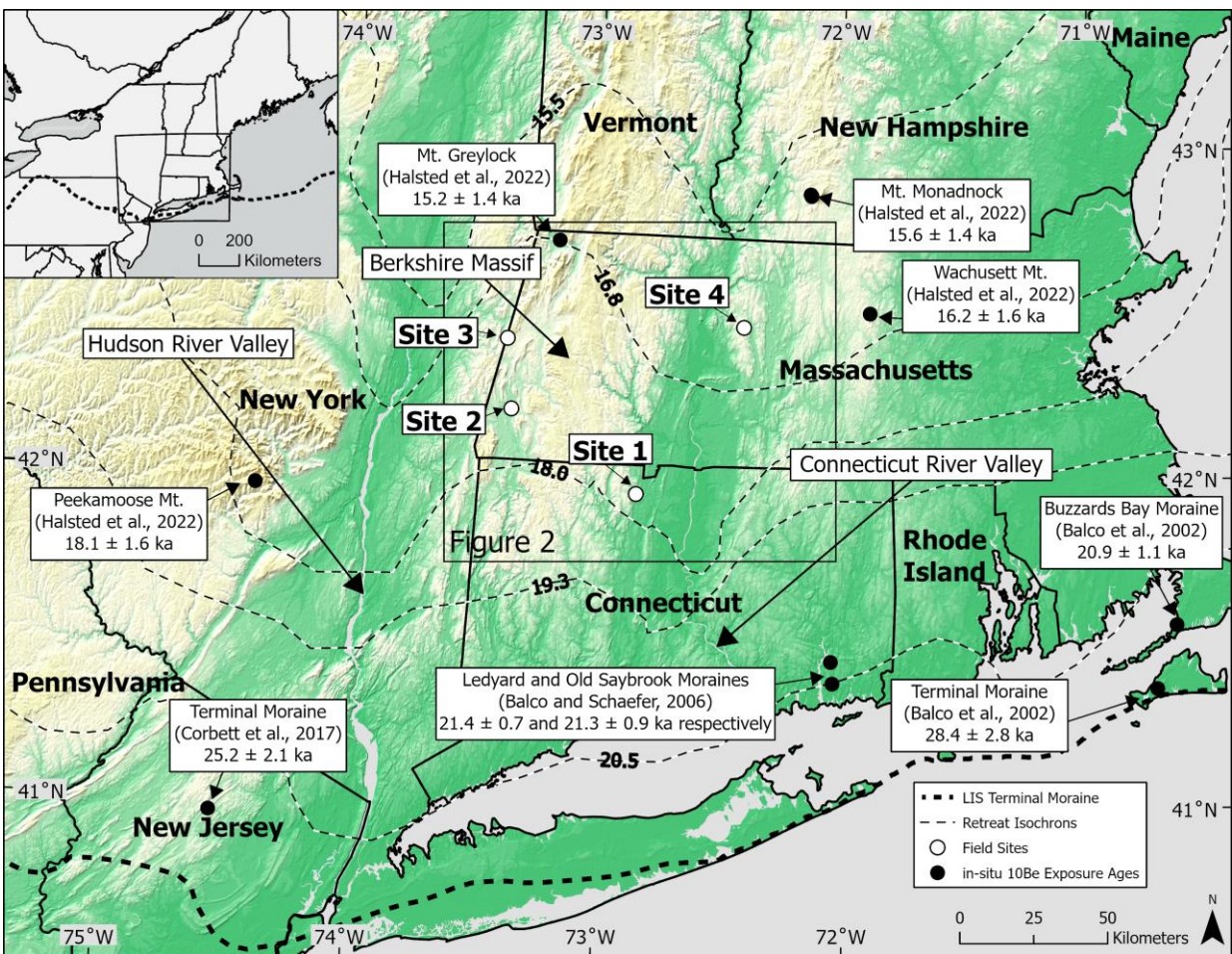

**Figure 1.** Map showing study location and regional topography in northeastern North America. Field sites from this study are marked (open white circles) as well as previous work in New England using in situ [10]Be exposure dating [1,7,14,16] (solid black circles; recalculated using LSDn scaling and global production rate where necessary). Thick dashed line represents the LIS terminal moraine as mapped by Dyke et al. [11]; thin dashed lines represent retreat isochrons during the last deglacial period, with estimated ages (ka) shown in bold numbers, from Dalton et al. [5]. Additional locations described in the text are labeled for regional context.

To understand better the history of the LIS retreat through northern Connecticut and Massachusetts, we determined the timing of deglaciation using 14 new in situ [10]Be exposure ages from 11 boulders and 3 bedrock samples. Our data, which fills an existing spatial gap, allow us to make inferences about the position of the LIS margins in central New England between 20 ka and 15 ka, coinciding with the onset of major deglaciation in the Northern Hemisphere [3,22].

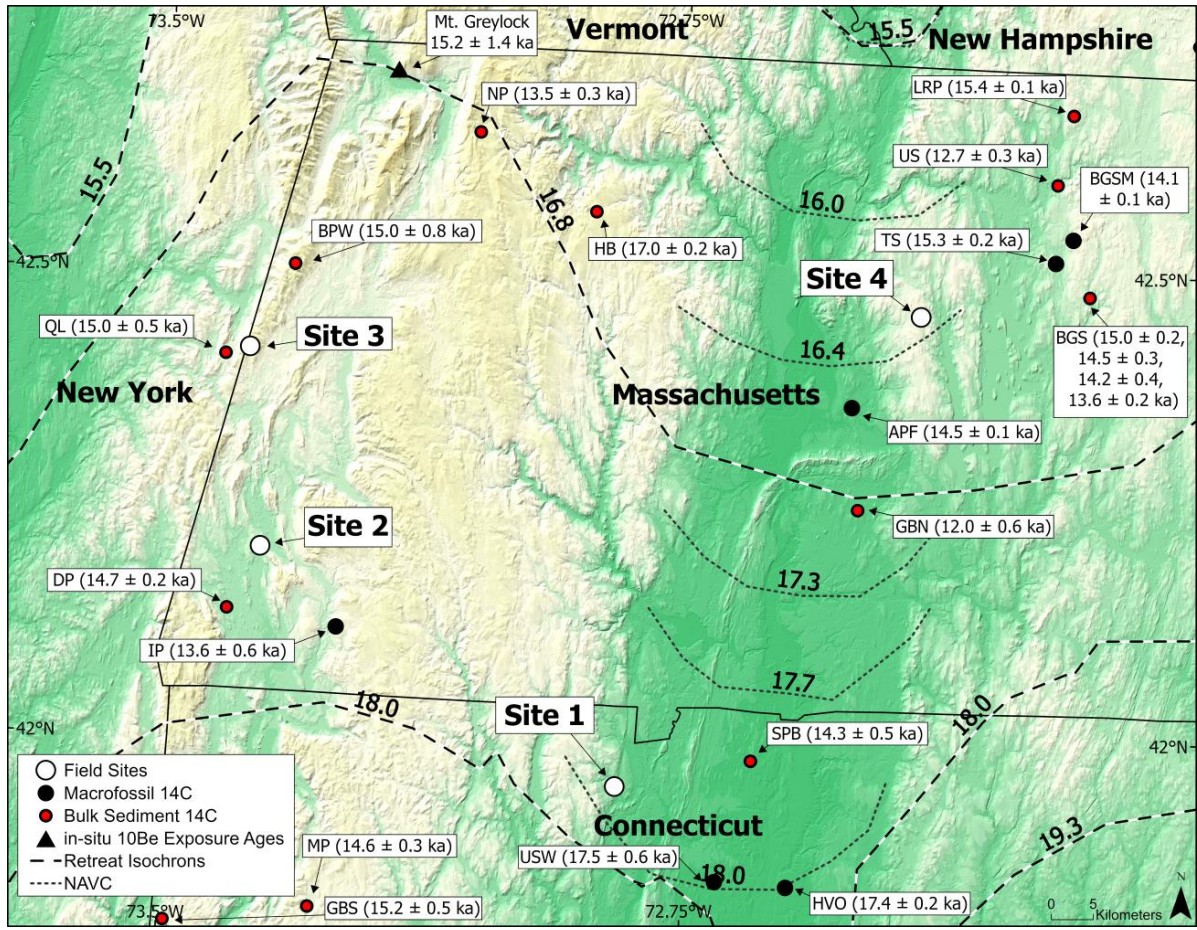

**Figure 2.** Map showing local topography around field sites (open white circles) and current deglacial constraints (in ka) based on Dalton et al. [5] (large black and white dashed lines) and Ridge et al. [12] (small black dashed lines). We identify all previously published, minimum-limiting deglacial organic [14]C ages (see Table 1, solid black circles and red circles with black outline), with abbreviated names corresponding to the table entries and [10]Be exposure ages (solid black triangle; Halsted et al. [16]) from the region.

**Table 1.** Compilation of previously published organic [14]C ages used to provide minimum limiting ages of deglaciation for the LIS in the study area.

| Location | Map Code | Latitude (°N) | Longitude (°W) | Elevation (m) | Material Type | [14]C Age and Uncertainty ([14]C yr BP) * | Calibrated Age and Uncertainty (yr BP) ** | Reference and Sample ID |
|---|---|---|---|---|---|---|---|---|
| Amherst, MA | APF | 42.360 | 72.510 | 43 | Terrestrial plant fragments | 12,370 ± 120 | 14,500 ± 120 | Rittenour [19], Beta-124780 |
| Black Gum Swamp, MA | BGS | 42.480 | 72.167 | 357 | Bulk Sediment | 12,400 ± 80 | 14,520 ± 270 | Anderson et al. [23], AA-40809 |
| Black Gum Swamp, MA | BGS | 42.480 | 72.167 | 357 | Bulk Sediment | 12,610 ± 80 | 15,000 ± 150 | Anderson et al. [23], AA-40812 |
| Black Gum Swamp, MA | BGS | 42.480 | 72.167 | 357 | Bulk Sediment | 11,690 ± 140 | 13,560 ± 160 | Foster and Zebryk [24], Beta-31366 |
| Black Gum Swamp, MA | BGS | 42.480 | 72.167 | 357 | Bulk Sediment | 12,240 ± 110 | 14,240 ± 380 | Anderson et al. [23], Beta-42117 |
| Black Gum Swamp, MA | BGSM | 42.542 | 72.192 | 358 | *Picea* fragments | 12,190 ± 60 | 14,100 ± 70 | Lindbladh et al. [18], Beta-192020 |
| Berry Pond, MA | BPW | 42.506 | 73.319 | 631 | Bulk Sediment | 12,680 ± 480 | 15,040 ± 750 | Whitehead [25], OWU-481 |
| Davis Pond, MA | DP | 42.136 | 73.408 | 213 | Bulk Sediment | 12,500 ± 50 | 14,700 ± 210 | Newby et al. [26,27], OS-55125 |
| Granby Bog, MA | GBN | 42.250 | 72.500 | 110 | Bulk Sediment | 10,300 + 370 | 12,020 ± 610 | Valastro et al. [26], TX-2946 |

**Table 1.** *Cont.*

| Location | Map Code | Latitude (°N) | Longitude (°W) | Elevation (m) | Material Type | $^{14}$C Age and Uncertainty ($^{14}$C yr BP) * | Calibrated Age and Uncertainty (yr BP) ** | Reference and Sample ID |
|---|---|---|---|---|---|---|---|---|
| Gross Bog, CT | GBS | 41.800 | 73.491 | 330 | Bulk Sediment | 12,750 ± 230 | 15,160 ± 500 | Newman et al. [28], RL-245 |
| Hawley Bog, MA | HB | 42.567 | 72.883 | 549 | Bulk Sediment | 14,000 ± 130 | 17,020 ± 220 | Bender et al. [29], WIS-1122 |
| Hitchcock Varve Outcrop, CT | HVO | 41.845 | 72.598 | 0 | Terrestrial plant leaves, mostly *Dryas integrifolia* | 14,300 ± 60 | 17,390 ± 150 | Ridge et al. [12], OS-77140 |
| Ivory Pond, MA | IP | 42.117 | 73.250 | 0 | *Picea glauca* cones | 11,630 ± 470 | 13,640 ± 570 | Moeller [17], GX-9259 |
| Little Royalston Pond, MA | LRP | 42.675 | 72.192 | 362 | Bulk Sediment | 12,910 ± 80 | 15,440 ± 130 | Oswald et al. [30], AA-58099 |
| Mohawk Pond, CT | MP | 41.817 | 73.283 | 360 | Bulk Sediment | 12,460 ± 110 | 14,630 ± 300 | Steventon and Kutzbach [31], WIS-1405 |
| North Pond, MA | NP | 42.650 | 73.053 | 586 | Bulk Sediment | 11,600 ± 280 | 13,500 ± 290 | Huvane and Whitehead [32], GX-4490 |
| Queechy Lake, NY | QL | 42.408 | 73.417 | 311 | Bulk Sediment | 12,680 ± 200 | 15,040 ± 450 | Stuiver [33], Y-2247 |
| Suffield Peat Bog, CT | SPB | 41.980 | 72.650 | 47 | Bulk Sediment | 12,200 ± 350 | 14,340 ± 540 | Rubin and Alexander [34], W-828 |
| Tom Swamp, MA | TS | 42.517 | 72.217 | 232 | Vascular plant macrofossils | 12,830 ± 120 | 15,330 ± 180 | Miller, [35], WIS-1210 |
| Unnamed Swamp, MA | US | 42.601 | 72.215 | 212 | Bulk Sediment | 10,800 ± 250 | 12,740 ± 300 | Rubin and Alexander [36], W-361 |
| Unnamed Pond, CT | USW | 41.850 | 72.700 | 0 | *Salix* wood fragments | 14,330 ± 430 | 174,70 ± 560 | Stone and Ashley [37], Beta-35211 |

* $^{14}$C ages (uncalibrated) as reported in source publications. BP = years before present (1950 AD). ** Calibrated years before present (calendar years before 1950 AD) were calculated using the MatCal software from Lougheed and Obrochta [38] and the IntCal20 calibration curve from Reimer et al. [39]. Uncertainties reported here are one half of the 68.2% probability distribution range of calibrated ages.

## 2. Background

### 2.1. Geographic Setting: Connecticut River and Hudson River Valleys

The study area encompasses sites in north-central Connecticut and west-central Massachusetts (Figure 2) within the Connecticut River and Hudson River Valleys that hosted LIS lobes and, later, large glacial lakes [12,40]. These valleys are separated by the Berkshire Massif [41], which is ~600 m higher than the Hudson River Valley and ~550 m higher than the Connecticut River Valley, possibly attributable to erosion-resistant metamorphic and igneous bedrock underlying the upland [42]. During deglaciation, the Hudson and Connecticut River Valleys constrained and channeled ice flow parallel to regional topography, resulting in south-flowing ice lobes from the LIS that persisted longer in the lowlands than in the highlands [5,11,12,16,43]. Upland areas, where bedrock is not exposed at the surface, are mantled by till [43], and often contain large boulders suitable for cosmogenic nuclide dating. In lowland areas, drainage basins trapped glacio-lacustrine sediment in proglacial lakes, allowing for the deposition of rhythmic glacial sediment, interpreted as varves [40,44,45].

### 2.2. In Situ Cosmogenic Nuclide Exposure Dating

The accumulation of cosmogenic nuclides in rocks exposed to cosmic rays at the Earth's surface is used to estimate exposure duration and therefore to determine the timing of past geomorphic events [46,47]. Spallation reactions occurring in minerals (for example, quartz) bombarded by cosmic rays result in the production of cosmogenic nuclides at and near the Earth's surface [48,49]. During ice occupation in formerly glaciated regions, cosmogenic nuclides do not accumulate due to the shielding of rock surfaces by overriding ice [50]. When the ice retreats and surfaces are exposed, cosmogenic nuclides accumulate at predictable rates [46]. Measuring the concentration of specific cosmogenic nuclides (most commonly $^{10}$Be) provides insight about the duration of exposure following ice retreat [1,14,49,51].

The use of cosmogenic nuclide dating to constrain the timing of glacial retreat depends on the assumptions of the material being dated. One assumption is subglacial erosion occurred deep enough to remove in situ $^{10}$Be produced during prior periods of exposure.

Where warm-based ice occupies the landscape for thousands of years, meters of erosion occur, reducing the [10]Be concentration in outcrops to low levels [50]. However, if ice is cold-based and non-erosive, and/or has a short period of occupation, erosion depth is limited and nuclides from previous exposure periods can remain in rocks [52,53]. If [10]Be from prior periods of exposure is not removed by erosion, calculated exposure ages will overestimate the true timing of deglaciation [51,54]. If boulders are disturbed (e.g., rolled) or shielded following exposure, and then subsequently exhumed, exposure ages will underestimate the timing of local deglaciation [55].

## 2.3. Previous Work: LIS Retreat Timing in New England

Early work to constrain ice retreat through New England relied on stratigraphic relationships and sedimentological observations [44,45,56–59]. With the advent of numerical dating methods, current constraints on deglaciation in most of Massachusetts and Connecticut now rely on glacial varve chronologies [12,44,45] and organic [14]C ages (from bulk sediment or macrofossil samples) from the bottom of lake or bog sediment cores [5,11,15]. Organic [14]C ages indicate the timing of post-glacial re-vegetation and are often interpreted as minimum limits on the timing of local deglaciation [60]. [14]C ages from both bulk sediment and macrofossil samples are used in New England deglacial chronologies, but age discrepancies between these methods have previously been noted [10,15]. Older bulk sediment ages are often attributed to incorporation in samples of carbon-bearing materials unrelated to deglaciation [61], and so younger [14]C ages from macrofossil samples are typically viewed as more accurately corresponding to the timing of post-glacial re-vegetation [15]. Both radiocarbon and varve-based chronologies suggest that ice retreated fully from Connecticut by 17.5–17.7 ka and from Massachusetts by 15.5 ka [5,12]. The North American Varve Chronology (NAVC) utilizes organic [14]C ages to anchor it to calendar years and Dalton et al.'s [5] chronology is based on the NAVC, adjusting the margins from Dyke's [11] isochrons and then extending them east and west of the Connecticut River Valley. More recent studies utilize in situ [10]Be exposure ages from erratic boulders, moraine boulders, or glacially scoured bedrock to estimate the timing of ice retreat; however, there are no published cosmogenic nuclide data in central New England [1,14,51,62–64].

## 2.4. Paleoclimate during LIS Retreat through New England

The global climate system changed significantly after the LGM. Summer insolation began increasing ~24 ka at high northern latitudes [13,65], increasing the duration and intensity of radiative forcing over the LIS [66]. Northward oceanic heat transport via the Gulf Stream was strong during the LGM but weakened significantly from 19 to 15 ka [67], reducing oceanic heat supply off the New England coast. Atmospheric $CO_2$ concentrations began increasing globally approximately 17 ka and were a major driver of Northern Hemisphere warming during deglaciation [68]. However, prior to 17 ka, global mean surface temperatures remained cold, particularly in regions surrounding the North Atlantic [13,66].

## 3. Study Sites

Our work focuses on four study sites within New England (Figure 1). We selected one site in Connecticut within the Connecticut River Valley Basin (Site 1) and three sites in Massachusetts: two in or near the Hudson River Valley (Sites 2 and 3) and one in the Connecticut River Valley (Site 4; Figure 2). All sites are above glaciolacustrine limits and are spatially separate, forming two N-S trending profiles allowing us to assess the LIS as it retreated northward at the end of the LGM. Additionally, sampling from two separate river valleys allows us to assess the dynamics of sub-lobes of the LIS.

### 3.1. Connecticut River Valley, Northern Connecticut (Site 1)

We collected samples from two schist bedrock outcrops and two boulders (one schist and one granite) near Broad Hill in West Granby, Connecticut (41°57.37′ N, 72°50.64′ W). Sampling occurred near a 20 m cliff face with scree scattered near the base. We sampled

boulders more than 500 m away from the cliff face to minimize the possibility that boulders were emplaced by rock fall off the cliff and instead are part of the surficial till. The landscape away from the cliff is hummocky and has abundant large rounded to subrounded boulders present at the surface (Figure 3). Intermittent bedrock outcrops of the Goshen formation (Devonian schist; [69]) are visible at the surface and contain abundant quartz veins that protrude 2–3 cm in positive relief.

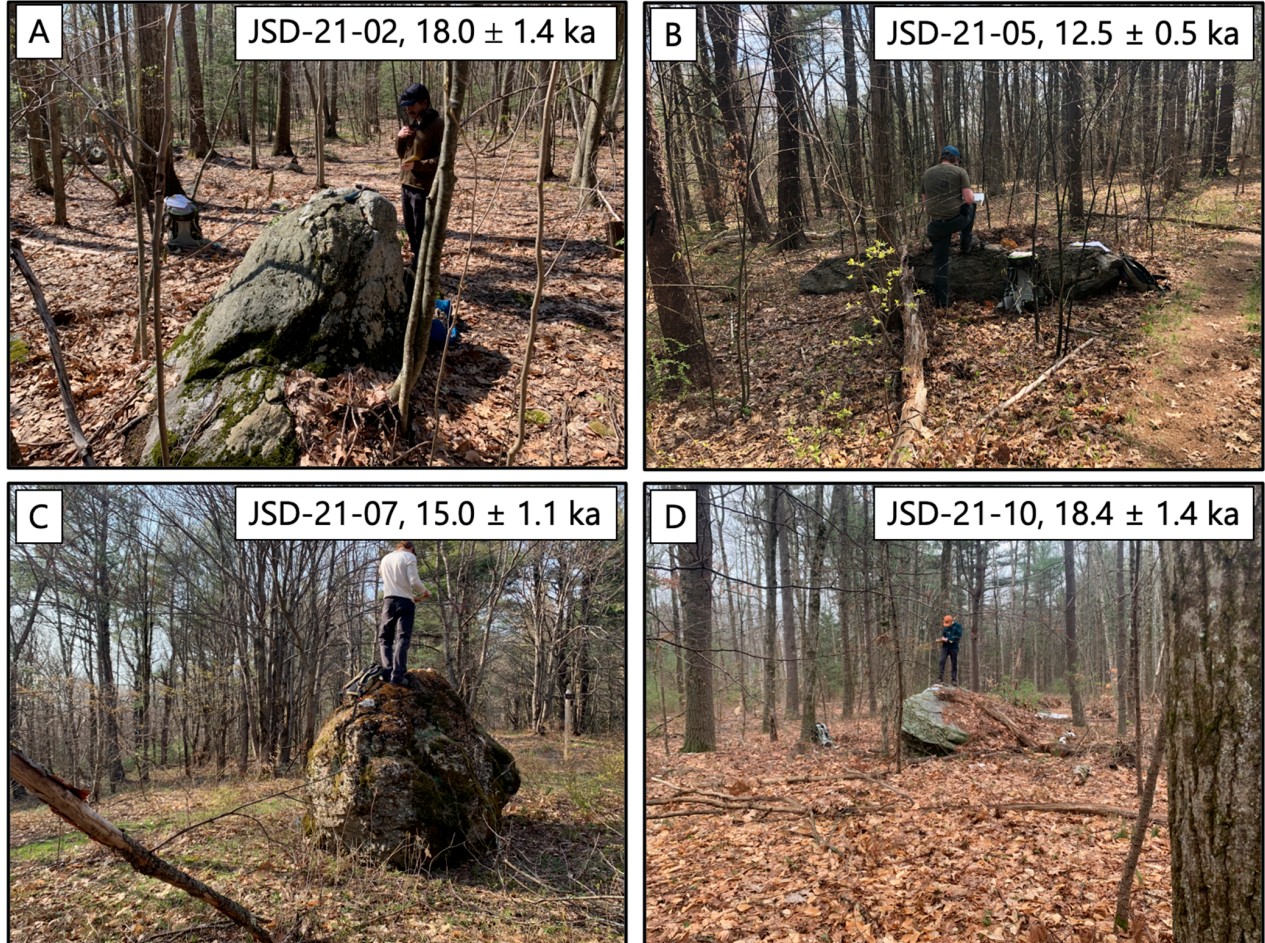

**Figure 3.** Representative samples from each study site showing the relatively flat deciduous forest landscape where we collected samples. (**A**) Connecticut River Valley in Northern Connecticut (Site 1; n = 4), (**B**) Housatonic Valley in Western Massachusetts (Site 2; n = 1), (**C**) Hudson River Valley drainage divide in Western Massachusetts (Site 3; n = 4), (**D**) Connecticut River Valley in Central Massachusetts (Site 4; n = 5). Labels show sample ID and exposure age ± external uncertainty.

*3.2. Housatonic Valley, Western Massachusetts (Site 2)*

We collected one sample from the quartz vein on top of a schist boulder near Lake Mansfield in Great Barrington, Massachusetts (42°12.206′ N, 73°22.069′ W) located in the Housatonic Valley of western Massachusetts (the watershed bordering the Hudson River Valley in the east). This site was selected due to the presence of the Great Barrington Boulder Train identified in 1910 by Frank Taylor [70]. The local topography is low relief (less than 50 m elevation change) with some boulders (5 m in diameter) scattered at the surface (Figure 3). The site is near the Housatonic River, but well above its floodplain, and boulders occur on a local topographic high, suggesting that no post-glacial movement following deposition occurred.

### 3.3. Hudson River Valley Drainage Divide, Western Massachusetts (Site 3)

We sampled one phyllite bedrock outcrop and three boulders (one phyllite and two quartz veins on schist) near Perry's Peak in Richmond, Massachusetts (42°24.823′ N, 73°22.771′ W). Located on a ridge of the Nassau Formation [71], the site is ~400–500 m above the Hudson River Valley to the west on the drainage divide with the Housatonic Valley ~200–300 m below to the east. Few boulders are present, despite the literature suggesting that this region was rich in boulders from the Richmond Boulder Train [70]. Boulders did not appear to represent the Nassau formation, so were likely erratic. However, this land was farmed extensively in the 18th and 19th centuries, as indicated by the presence of stone walls. At that time, farmers may have moved the boulders. Few bedrock outcrops were present, mostly at higher elevations.

### 3.4. Connecticut River Valley, Central Massachusetts (Site 4)

We sampled five boulders near Shutesbury, Massachusetts (42°27.472′ N, 72°24.694′ W). The study site is on the upland between the Ashuelot and Chicopee River watersheds, both of which drain into the Connecticut River Valley. This location was above the paleo-shoreline of Glacial Lake Hitchcock. Large boulders are scattered across the area, some in very close proximity to each other; all are coarse-grained gneiss with large quartz veins (Figure 3). The terrain is hummocky, possibly due to kame and kettle terrain formed during ice retreat.

## 4. Methods

### 4.1. Field Sampling

We sampled boulders (n = 11) and bedrock (n = 3) from the four sites in April 2021 (Table 2). Sub-rounded boulders, assumed to be glacially entrained, were selected if they showed no indication of post-glacial movement, shielding, or sub-aerial erosion. We selected bedrock in locations near boulders and only where there was no evidence of post-glacial shielding by soil or sediment. We removed the top few centimeters of rock with a hammer and chisel. We recorded site parameters for $^{10}$Be production rate estimation, including latitude, longitude, elevation, rock surface strike and dip at the sampled location, and the azimuth and elevation of local topography for shielding calculations using the online topographic shielding calculator described in Balco et al. [72].

**Table 2.** Sample location and field data for 14 boulder and bedrock samples.

| Sample Name | Type | Site | Drainage Basin | Latitude (°N) * | Longitude (°W) * | Elevation (m a.s.l.) * | Sample Thickness (cm) ** | Shielding Factor † | Rock Type | Boulder Dimensions (m) †† |
|---|---|---|---|---|---|---|---|---|---|---|
| JSD-21-01 | Bedrock | 1 | Connecticut | 41.94922 | 72.84601 | 181 | 2.5 | 0.999 | Schist | N/A |
| JSD-21-02 | Boulder | 1 | Connecticut | 41.95058 | 72.84819 | 199 | 4.0 | 0.981 | Schist | 2 × 0.75 × 1.25 |
| JSD-21-03 | Bedrock | 1 | Connecticut | 41.95093 | 72.84751 | 210 | 2.5 | 1.000 | Schist | N/A |
| JSD-21-04 | Boulder | 1 | Connecticut | 41.95080 | 72.84513 | 155 | 4.0 | 0.902 | Granite | 2.2 × 1.5 × 1.5 |
| JSD-21-05 | Boulder | 2 | Hudson | 42.20230 | 73.36130 | 281 | 1.0 | 1.000 | Quartz Vein | 4 × 2 × 0.75 |
| JSD-21-06 | Bedrock | 3 | Hudson | 42.41372 | 73.37952 | 582 | 3.0 | 0.996 | Phyllite | N/A |
| JSD-21-07 | Boulder | 3 | Hudson | 42.41576 | 73.38173 | 581 | 3.0 | 0.997 | Quartz Vein | 2.5 × 2 × 2.25 |
| JSD-21-08 | Boulder | 3 | Hudson | 42.41084 | 73.37003 | 495 | 1.5 | 0.997 | Phyllite | 2.25 × 1.25 × 1 |
| JSD-21-09 | Boulder | 3 | Hudson | 42.40790 | 73.37010 | 463 | 5.5 | 0.948 | Quartz Vein | 1 × 0.5 × 0.75 |
| JSD-21-10 | Boulder | 4 | Connecticut | 42.45786 | 72.41156 | 374 | 4.0 | 0.999 | Gneiss | 2.5 d × 2 × 1.5 |
| JSD-21-11 | Boulder | 4 | Connecticut | 42.45801 | 72.41139 | 374 | 3.0 | 0.991 | Gneiss | 2 × 1.5 × 1.5 |
| JSD-21-12 | Boulder | 4 | Connecticut | 42.45805 | 72.41141 | 371 | 1.5 | 0.999 | Gneiss | 3 × 1.75 × 1 |
| JSD-21-13 | Boulder | 4 | Connecticut | 42.45845 | 72.41609 | 348 | 1.0 | 0.996 | Gneiss | 2.75 × 1.25 × 2 |
| JSD-21-14 | Boulder | 4 | Connecticut | 42.45902 | 72.41552 | 354 | 2.0 | 0.997 | Gneiss | 2 × 1 × 1 |

* Latitude, Longitude and Elevation were all measured using the GPS app "Gaia GPS" on an iPhone. Latitude and Longitude were recorded using decimal degrees and the WGS 84 datum. Elevation was recorded in feet. We allowed the app to stabilize for at least 5 min before recording the values. ** Sample thickness was measured using a metric ruler, averaging the thickness from multiple points on each sample. † We recorded the azimuth and inclination of all prominent topographic features above the horizon in a 360° radius around each sample location using the inclinometer feature on the "Rockd" app on an iPhone. The shielding factor was calculated from these values in the online topographic shielding calculator described in Balco et al. [72]. †† Reported as the length × width × height.

## 4.2. Sample Preparation and Measurement

We measured the average sample thickness for each sample, and then isolated quartz at the University of Vermont according to methods described in Kohl and Nishiizumi, [73]. We verified quartz purity using a Perkin Elmer, Avio 200 Inductively Coupled Plasma Optical Emission Spectrometer. We isolated and purified beryllium at the National Science Foundation/University of Vermont Community Cosmogenic Facility with the methods described in Corbett et al. [74]. We prepared samples in two separate batches, each including one blank and one liquid reference material [75]. We digested between 17.5 and 21.1 g of quartz and added 250 μg $^9$Be to each using an in-house diluted carrier, termed UVM-SPEX, created from the dilution of SPEX 10,000 ppm Be standard, with a resulting concentration of 304 μg mL$^{-1}$ (Table 3).

**Table 3.** Sample preparation and laboratory information for $^{10}$Be/$^9$Be analysis.

| Sample Name | Quartz Mass (g) | Mass of $^9$Be Added (μg) * | AMS Cathode Number | Measured $^{10}$Be/$^9$Be Ratio ** | Measured $^{10}$Be/$^9$Be Ratio Uncertainty ** | Background-Corrected $^{10}$Be/$^9$Be Ratio † | Background-Corrected $^{10}$Be/$^9$Be Ratio Uncertainty † | $^{10}$Be Concentration ($10^4$ atoms g$^{-1}$) | $^{10}$Be Concentration Uncertainty ($10^3$ atoms g$^{-1}$) |
|---|---|---|---|---|---|---|---|---|---|
| JSD-21-01 | 21.118 | 250.5 | 163707 | $1.01 \times 10^{-13}$ | $5.54 \times 10^{-15}$ | $9.79 \times 10^{-14}$ | $5.59 \times 10^{-15}$ | 7.76 | 4.43 |
| JSD-21-02 | 21.082 | 250.6 | 163708 | $9.90 \times 10^{-14}$ | $4.70 \times 10^{-15}$ | $9.56 \times 10^{-14}$ | $4.75 \times 10^{-15}$ | 7.59 | 3.77 |
| JSD-21-03 | 20.911 | 250.2 | 163709 | $1.06 \times 10^{-13}$ | $7.27 \times 10^{-15}$ | $1.02 \times 10^{-13}$ | $7.31 \times 10^{-15}$ | 8.17 | 5.84 |
| JSD-21-04 | 20.911 | 250.5 | 163710 | $1.10 \times 10^{-13}$ | $4.67 \times 10^{-15}$ | $1.06 \times 10^{-13}$ | $4.72 \times 10^{-15}$ | 8.51 | 3.77 |
| JSD-21-05 | 17.477 | 249.4 | 163711 | $6.56 \times 10^{-14}$ | $3.15 \times 10^{-15}$ | $6.22 \times 10^{-14}$ | $3.22 \times 10^{-15}$ | 5.93 | 3.07 |
| JSD-21-06 | 20.702 | 249.6 | 163712 | $1.21 \times 10^{-13}$ | $4.38 \times 10^{-15}$ | $1.17 \times 10^{-13}$ | $4.43 \times 10^{-15}$ | 9.46 | 3.57 |
| JSD-21-07 | 20.818 | 249.9 | 163713 | $1.18 \times 10^{-13}$ | $4.27 \times 10^{-15}$ | $1.15 \times 10^{-13}$ | $4.33 \times 10^{-15}$ | 9.21 | 3.47 |
| JSD-21-08 | 21.003 | 250.2 | 163715 | $5.19 \times 10^{-14}$ | $3.71 \times 10^{-15}$ | $4.85 \times 10^{-14}$ | $3.78 \times 10^{-15}$ | 3.86 | 3.01 |
| JSD-21-09 | 21.040 | 250.5 | 163716 | $5.92 \times 10^{-14}$ | $3.34 \times 10^{-15}$ | $5.58 \times 10^{-14}$ | $3.41 \times 10^{-15}$ | 4.44 | 2.71 |
| JSD-21-10 | 20.900 | 249.9 | 163717 | $1.22 \times 10^{-13}$ | $5.32 \times 10^{-15}$ | $1.18 \times 10^{-13}$ | $5.36 \times 10^{-15}$ | 9.46 | 4.28 |
| JSD-21-11 | 20.877 | 249.2 | 163718 | $1.11 \times 10^{-13}$ | $4.13 \times 10^{-15}$ | $1.08 \times 10^{-13}$ | $4.19 \times 10^{-15}$ | 8.59 | 3.34 |
| JSD-21-12 | 21.012 | 251.2 | 163719 | $1.21 \times 10^{-13}$ | $4.36 \times 10^{-15}$ | $1.18 \times 10^{-13}$ | $4.42 \times 10^{-15}$ | 9.41 | 3.53 |
| JSD-21-13 | 20.971 | 250.2 | 163720 | $1.04 \times 10^{-13}$ | $3.76 \times 10^{-15}$ | $1.01 \times 10^{-13}$ | $3.82 \times 10^{-15}$ | 8.03 | 3.05 |
| JSD-21-14 | 20.905 | 249.8 | 163721 | $1.16 \times 10^{-13}$ | $5.28 \times 10^{-15}$ | $1.12 \times 10^{-13}$ | $5.33 \times 10^{-15}$ | 8.97 | 4.25 |

* $^9$Be was added from an in-house diluted carrier, termed UVM-SPEX, created from dilution of SPEX 10,000 ppm Be standard, with a resulting concentration of 304 μg mL$^{-1}$. ** Isotopic analysis was conducted at PRIME Laboratory; ratios were normalized against standard 07KNSTD3110 with an assumed ratio of $2.850 \times 10^{-12}$ [76]. † Measured ratios were corrected for backgrounds using a single blank prepared with and analyzed with the samples. Background-corrected uncertainties include sample measurement uncertainty and blank uncertainty propagated in quadrature. Blank ratio: $3.40 \pm 0.68 \times 10^{-15}$.

Accelerator Mass Spectrometer (AMS) measurements of $^{10}$Be/$^9$Be were performed at Purdue Rare Isotope Measurement (PRIME) Laboratory. Samples were normalized to the primary standard of 07KNSTD3110 with an assumed $^{10}$Be/$^9$Be ratio of $2.850 \times 10^{-12}$ [76]. Measured sample ratios ranged from $5.19 \times 10^{-14}$ to $12.2 \times 10^{-14}$. We corrected for backgrounds by subtracting the blank ratio ($3.40 \pm 0.68 \times 10^{-15}$) from the sample ratios and propagating uncertainties in quadrature.

## 4.3. Age Calculation

Exposure age estimates were calculated using Version 3 of the online exposure age calculator described in Balco et al. [72], calculated with the global production rate from Borchers et al. [77], using the LSDn scaling scheme from Lifton et al. [78], and assuming a quartz density of 2.65 g cm$^{-3}$. We selected the global production rate because we are comparing ages to organic $^{14}$C dates and the North American Varve Chronology, both of which were used to calibrate the Northeastern North America production rate [79]. Exposure ages assume no nuclide inheritance from previous exposure and that no shielding or erosion occurred since deglaciation. We assessed for outliers using the iceTEA "Remove Outliers" tool from Jones et al. [80].

### 5. Results

Cosmogenic exposure ages range from 12.5 ± 0.7 ka to 22.4 ± 1.7 ka (Table 4; Figures 4 and 5; internal uncertainties). Given independent constraints on the LIS margin reaching northern New England by at least 14 ka, (e.g., [12,16,51,62,81–83]), we exclude three implausibly young values (JSD-21-07 from Site 2, and JSD-21-08 and JSD-21-09 from Site 3; Table 4; Figure 5). These boulders may have been disturbed following glacial retreat by widespread anthropogenic landscape change that occurred in the 17th–20th centuries in New England and resulted in the movement and removal of boulders to clear fields for agriculture [84]. In addition, these boulders may have experienced partial shielding by soil thus causing measured ages to record processes other than simple exposure.

**Table 4.** Calculated exposure ages based on in situ $^{10}$Be concentrations.

| Sample Name * | Type | Site | $^{10}$Be Exposure Age (ka) $^{†}$ | $^{10}$Be Internal Uncertainty (ka) | $^{10}$Be External Uncertainty (ka) |
|---|---|---|---|---|---|
| JSD-21-01 | Bedrock | 1 | 18.1 | 1.0 | 1.5 |
| JSD-21-02 | Boulder | 1 | 18.0 | 0.9 | 1.4 |
| JSD-21-03 | Bedrock | 1 | 18.5 | 1.3 | 1.7 |
| *JSD-21-04* | *Boulder* | *1* | *22.4* | *1.0* | *1.7* |
| *JSD-21-05* | *Boulder* | *2* | *12.5* | *0.7* | *1.0* |
| JSD-21-06 | Bedrock | 3 | 15.4 | 0.6 | 1.1 |
| JSD-21-07 | Boulder | 3 | 15.0 | 0.6 | 1.1 |
| *JSD-21-08* | *Boulder* | *3* | *6.7* | *0.5* | *0.7* |
| *JSD-21-09* | *Boulder* | *3* | *8.5* | *0.5* | *0.7* |
| JSD-21-10 | Boulder | 4 | 18.4 | 0.8 | 1.4 |
| JSD-21-11 | Boulder | 4 | 16.8 | 0.7 | 1.2 |
| JSD-21-12 | Boulder | 4 | 18.1 | 0.7 | 1.3 |
| JSD-21-13 | Boulder | 4 | 15.8 | 0.6 | 1.1 |
| JSD-21-14 | Boulder | 4 | 17.6 | 0.8 | 1.3 |

* Samples in italics are considered outliers and are not included in this analysis. $^{†}$ Ages are calculated using Version 3 of the online exposure age calculator described in Balco et al. 2008 [72], calculated using the global production rate [77] and LSDn scaling scheme [78]. Internal uncertainty propagates only AMS analytical error. External uncertainty includes uncertainty in production rate calibration and altitude/latitude scaling.

The average exposure age for Site 1 in the Connecticut River Valley near the Connecticut and Massachusetts border is 18.3 ± 0.3 ka (n = 3; mean ± 1 SD; Figure 5). Bedrock (18.3 ± 0.3 ka, n = 2; mean ± 1 SD) and boulder (20.2 ± 3.1 ka, n = 2) samples agree within uncertainties. According to a two-tailed generalized extreme Studentized deviate test [80], sample JSD-21-04 was identified as an outlier at Site 1. This may be explained by inheritance from previous exposure; however, due to the small sample size, it is difficult to confidently discard this sample as an outlier; if it is included, the average exposure age for Site 1 becomes 19.3 ± 2.1 ka (n = 4; mean ± 1 SD). For Site 2 in western Massachusetts, there is a single 12.5 ± 0.7 ka (internal uncertainty) sample (Table 4; Figure 4). At Site 3 in Western Massachusetts north of Site 2, the average age of samples is 15.2 ± 0.3 ka (n = 2). Site 4 in the Connecticut River Valley has an average exposure age of 17.4 ± 1.0 ka (n = 5).

The distributions of ages in the Connecticut and Hudson River Valleys appear to differ from one another. Samples from northern Connecticut (Site 1) suggest exposure by 18.3 ± 0.3 ka. Samples from Central Massachusetts, 60 km north (Site 4), suggest exposure by at least 17.4 ± 1.0 ka (Figure 2). Samples from Site 3 in the Hudson River Valley suggest exposure occurred 15.2 ± 0.3 ka, which is approximately 2 kyr later than at Site 4 at a similar latitude in the Connecticut River Valley (Figures 2 and 5).

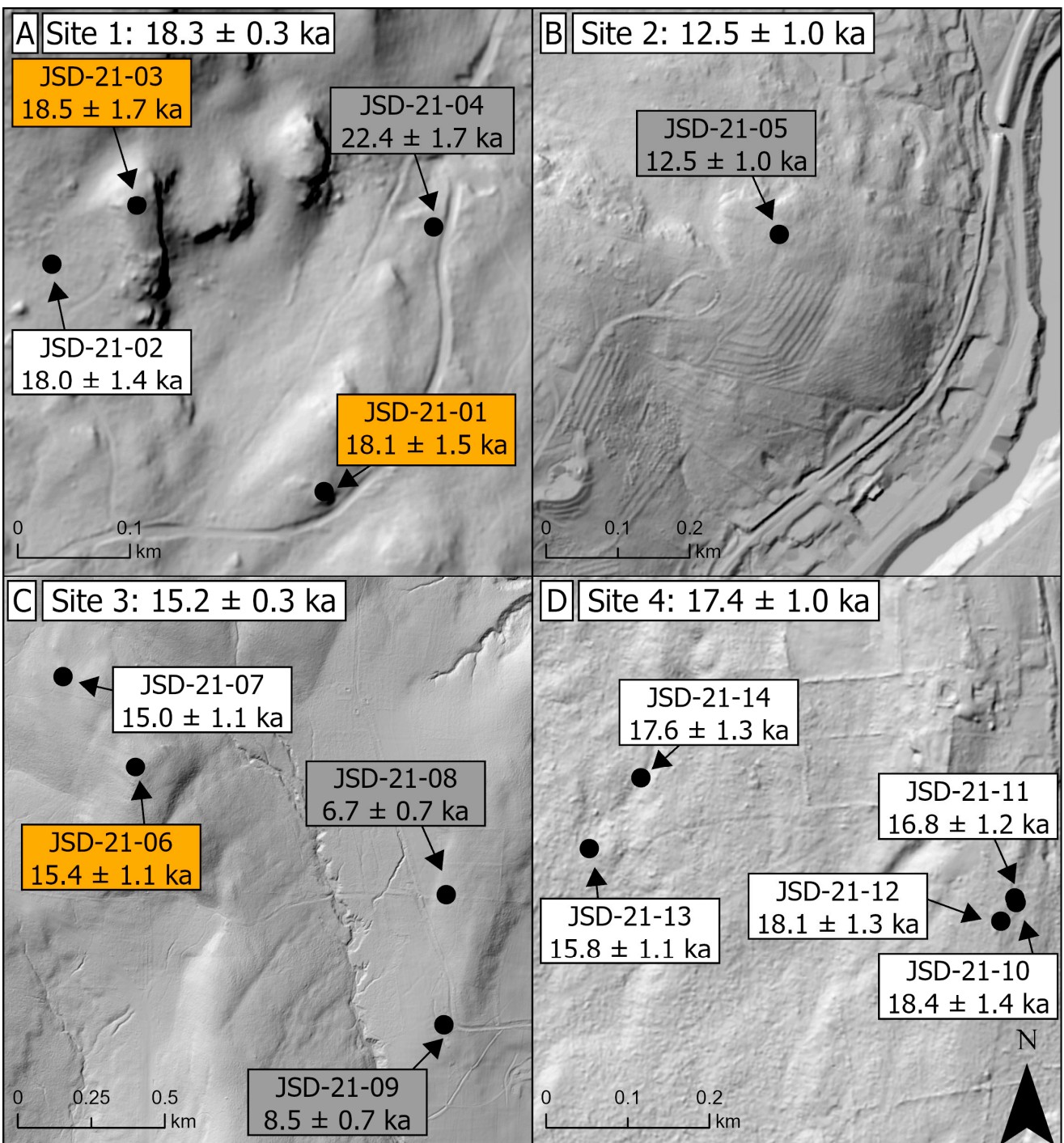

**Figure 4.** Map of each of four field sites showing sample locations overlain on 1 m LiDAR hillshade. Calculated exposure age ± external uncertainty for each sample and average of accepted samples ± 1 SD for each site. Samples with orange highlight are bedrock, all other samples are boulders. Samples that we consider outliers are indicated by a grey background. (**A**) Site 1—Northern Connecticut in the Connecticut River Valley, (**B**) Site 2—Western Massachusetts in the Housatonic River Valley, (**C**) Site 3—Western Massachusetts on the Hudson River Valley drainage divide, (**D**) Site 4—Central Massachusetts in the Connecticut River Valley.

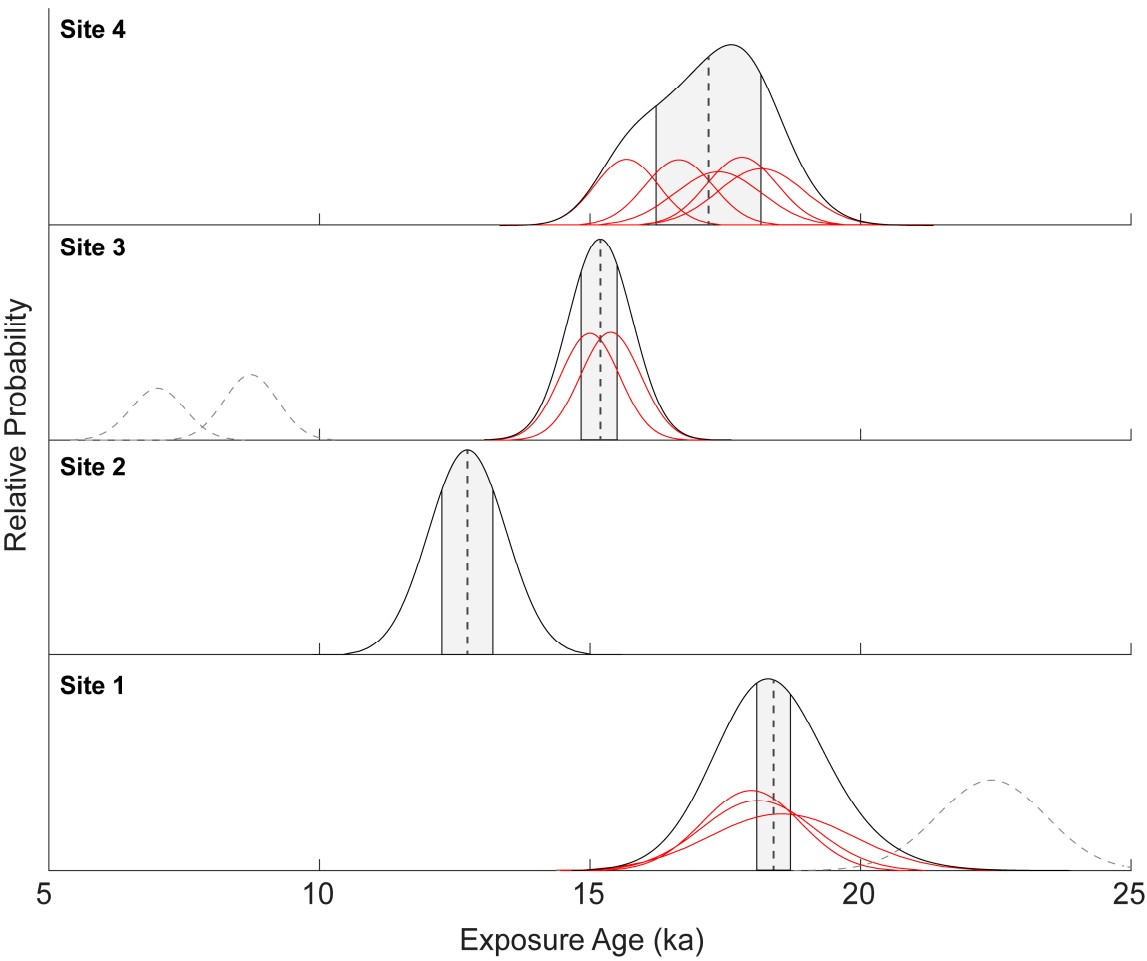

**Figure 5.** Kernel density functions of exposure ages for each site in this study (black). Probability density functions for individual [10]Be ages are plotted in red. Ages identified as outliers are plotted as gray dashed lines.

## 6. Discussion

### 6.1. Regional Significance of Exposure Ages

Exposure ages from study sites in the Connecticut River Valley decrease from south to north (Figure 5), but deglaciation in the central Connecticut River Valley ($17.4 \pm 1.0$ ka, n = 5, Site 1) may have occurred earlier than in the Hudson River Valley at a similar latitude ($15.2 \pm 0.3$, n = 2, Site 3). The small number of samples at each site limits our ability to reliably constrain retreat timing differences between the two lobes. Samples in the Connecticut River Valley support deglaciation to the border of Connecticut and Massachusetts by $18.3 \pm 0.3$ ka (Site 1, n = 3), with the LIS margin reaching central Massachusetts by $17.4 \pm 1.0$ ka (Site 1, n = 5). Given the uncertainty on ages for these two sites, our ages suggest rapid ice sheet retreat through this area. Other reconstructions suggest that ice retreated at approximately the same time through these valleys [5,12].

The sample sites we explore here are bracketed in southern Connecticut by the cosmogenic exposure ages from the Ledyard ($21.4 \pm 0.7$ ka) and Old Saybrook ($21.3 \pm 0.9$ ka) moraines [1] and Mt. Greylock in northern Massachusetts ($15.2 \pm 1.4$ ka) [16]; hence, our data fill in the previously existing gap in the [10]Be exposure age chronology in New England (Figure 6). Considered together, the cosmogenic data suggest continuous and likely rapid ice sheet retreat through Connecticut and Massachusetts between 20 and 15 ka, a time when persistently cold stadial conditions existed in the region.

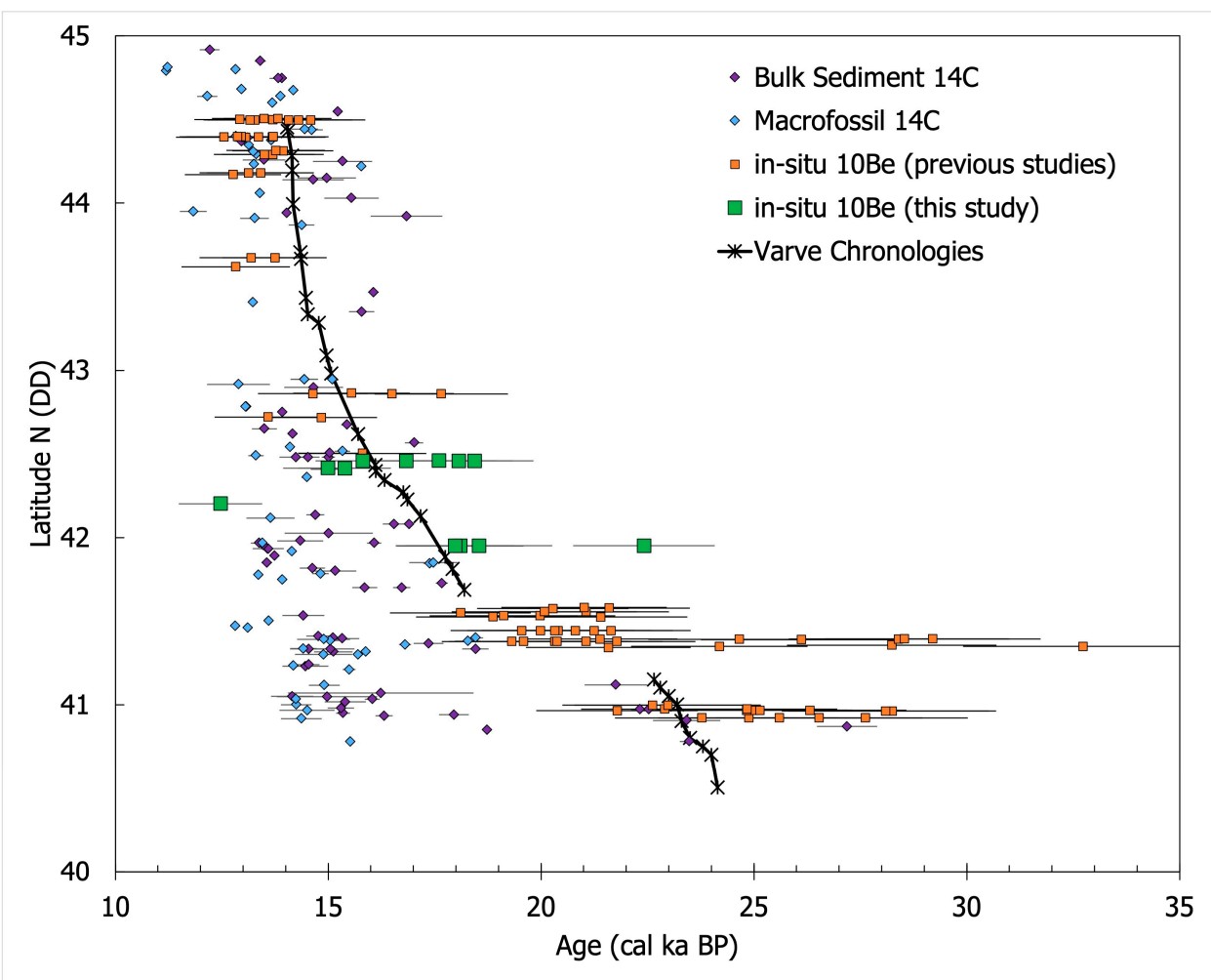

**Figure 6.** Plot of age versus latitude for all samples corresponding to LIS deglaciation in New England. New ages for this study are in green and fill a conspicuous hole in the previously published $^{10}$Be chronology (ages for JSD-21-08 and -09 are not shown). $^{10}$Be exposure ages agree better with the varve chronology than organic radiocarbon ages. The results of all three chronometers converge to the north. All $^{14}$C values are recalibrated using the same method as described in Table 1.

*6.2. Comparison to Other Regional LIS Retreat Chronologies*

Our findings are consistent with the NAVC; both records suggest similar timing for deglaciation and thus the rate of LIS retreat in the Connecticut River Valley. At the southernmost site (1), the ice retreated by 18.3 ± 0.3 ka and reached the northernmost site (~60 km distance, Site 4) at approximately 17.4 ± 1.0 ka. This chronology is similar to, but slightly older than, that suggested by the NAVC, which places ice retreat from northern Connecticut by 18.0 ka and central Massachusetts by 16.4 ka, within the uncertainty of our calculated values. Ice retreat rates estimated from exposure ages are similar to those estimated by the NAVC estimates; we calculate a most-likely retreat rate of 67 m y$^{-1}$ north (range: 27–>100 m y$^{-1}$) through the Connecticut River Valley; the range estimated by Ridge et al. [12] through this same region is 30–40 m y$^{-1}$. Our choice of $^{10}$Be production rate calibration data set (Section 4.3) could explain why the most-likely exposure ages are slightly older than the ice retreat dates suggested by the NAVC, as could $^{10}$Be inherited from prior periods of exposure. The global calibration data set that we use here [77] implies slightly lower $^{10}$Be production rates (~4%), and thus older calculated exposure ages (~0.7 kyr), for central New England compared to a regional calibration data set based on the NAVC ([79]; with LSDn scaling).

Although these new cosmogenic ages align reasonably with NAVC-based retreat dates, they do not always agree well with existing core bottom radiocarbon ages (Figures 2 and 6). Near Site 1 in the Connecticut River Valley near the Connecticut-Massachusetts border, the oldest calibrated radiocarbon ages from macrofossils are $17.5 \pm 0.6$ ka (USW) and $17.4 \pm 0.2$ ka (HVO), nearly 1 ka younger than exposure ages slightly farther north (Table 1; Figure 2). At Site 4 in central Massachusetts, the average exposure age is $17.4 \pm 1.0$ ka, but nearby radiocarbon samples are $14.5 \pm 0.1$ ka (APF; macrofossil; Figure 2), $15.3 \pm 0.2$ ka (TS; macrofossil; Figure 2), and $15.0 \pm 0.2$ ka (BGS; bulk sediment; Figure 2), or 2–3 ka younger. Despite the differences in deglacial ages between chronologies in the Connecticut River Valley, radiocarbon ages agree better at Site 3 in Western Massachusetts. Two radiocarbon ages from bulk sediment nearby are $15.0 \pm 0.8$ ka (BPW; Table 1; Figure 2) and $15.0 \pm 0.5$ (QL; Table 1; Figure 2) agreeing with the $15.2 \pm 1.6$ ka cosmogenic estimate of ice retreat there. Delayed revegetation of the landscape following the initial LIS retreat may account for the larger differences between radiocarbon dates and new exposure ages in the older Connecticut River Valley site (e.g., [10]).

*6.3. Possible Paleoclimate Forcings on LIS retreat*

The timing of LIS retreat through Connecticut and Massachusetts between 18.3 and 15.2 ka is coincident with a cold stadial period across the Northern Hemisphere [13]. Existing paleoclimate reconstructions suggest that prior to 17 ka, rising summer insolation was the primary driver of deglaciation [13,66]. Some suggest that LIS retreat during a cold period is unlikely and thus conclude, based on radiocarbon ages, that LIS retreat did not begin until approximately 16 ka [15]. However, the change in solar insolation starting at 24 ka [65] has been linked to LIS recession elsewhere along the southern margin [66]. Following the initial retreat of the LIS, the Gulf Stream slowed significantly beginning at approximately 19 ka, bringing less warmth from the equatorial regions to the North Atlantic. Despite this, we measure exposure ages from northern Connecticut at approximately 18.3 ka, prior to the onset of major global deglaciation at 17 ka due to rising $CO_2$. Additionally, we find exposure ages from central Massachusetts, 60 km north, 2 kyr later, suggesting that during this period the LIS in New England was actively retreating. Retreat during this time is supported by the NAVC. Together, these observations suggest that rising summer insolation, identified elsewhere as being responsible for LIS retreat [66], may be a cause for margin retreat in central New England as well, given that we date retreat prior to the re-strengthening of the Gulf Stream and rising $CO_2$ levels.

## 7. Conclusions

Constraining the timing of deglaciation of the LIS in central New England is important for understanding the co-evolution of regional climate with deglaciation at the end of the Last Glacial Maximum. Fourteen new in situ $^{10}$Be exposure ages from Northern Connecticut and Massachusetts fill a conspicuous gap in cosmogenic dates of LIS retreat and allow a better understanding of the LIS and climate co-evolution at the end of the LGM. We determined that ice in the Connecticut River Valley retreated from Connecticut by approximately 18.3 ka and retreated ~60 km north through central Massachusetts by 17.4 ka. In western Massachusetts, ice from the Hudson Valley lobe reached a similar latitude by 15.2 ka. These data suggest that the LIS margin in central New England continued retreating during a stadial period when the Gulf Stream was weak, and the Northern Hemisphere was cold.

**Author Contributions:** Conceptualization, J.S.D., C.T.H. and P.R.B.; methodology, L.B.C. and P.R.B.; formal analysis, J.S.D. and C.T.H.; investigation, J.S.D., C.T.H., L.B.C. and M.W.C.; resources, L.B.C., P.R.B. and M.W.C.; data curation, C.T.H.; writing—original draft preparation, J.S.D.; writing—review and editing, C.T.H., L.B.C., P.R.B. and M.W.C.; visualization, J.S.D. and C.T.H.; supervision, C.T.H., L.B.C. and P.R.B.; project administration, P.R.B.; funding acquisition, J.S.D. and P.R.B. All authors have read and agreed to the published version of the manuscript.

**Funding:** This research was funded by the University of Vermont (UVM) Geology Department, Hawley Mudge grant; UVM College of Arts and Sciences, APLE Award; the Geological Society of America Northeastern Section, Stephen G. Pollock Undergraduate Student Research Grant; UVM Summer Undergraduate Research Fellowship, Sustainability Summer Fellowship; and by funding provided to P.R.B. by the National Science Foundation under NSF-EAR-1735676 and NSF-EAR-1602280.

**Data Availability Statement:** Data associated with this project will be made available on the ICE-D: Laurentide database upon publication of this work. ICE-D can be accessed at ice-d.org.

**Acknowledgments:** We thank Al Werner for his assistance finding suitable boulders to sample in Western Massachusetts.

**Conflicts of Interest:** The authors declare no conflict of interest. The funders had no role in the design of the study; in the collection, analysis, or interpretation of the data; in the writing of the manuscript; or in the decision to publish the results.

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
