# Peer review of "In Situ Cosmogenic 10Be Dating of Laurentide Ice Sheet Retreat from Central New England, USA"

_geosciences, doi:10.3390/geosciences13070213_

Round 1

Reviewer 1 Report

In this study, Drebber et al. explore the deglaciation history of the Laurentide Ice Sheet and focus on the retreat in New England, USA. Our knowledge of the response of the ice sheets to climate fluctuations in the past is crucial in understanding the current and future response of the Antarctic and Greenlandic ice sheets to global warming and the estimation/modeling of severe consequences of fast loss of global ice volume. In this context, this study reveals a missing piece of the deglaciation history of the Laurentide Ice Sheet, as excellently shown in Figure 6 in the manuscript. The authors use surface exposure dating with cosmogenic 10Be to reconstruct the chronology of the deglaciation in central New England. They provide 14 new surface exposure ages from 4 sites and present a regional deglaciation history by reconciling these with the existing surface exposure and radiocarbon ages. Overall, I think that the manuscript is well written. The methodology applied and data collection is appropriate for the purpose of the study. The results of the study are robust. The discussion chapter reads also well, however, a little bit short and concise. There is room to extend the discussed topics. However, after a minor revision, I think this manuscript deserves to be published in Geosciences. It will have a long shelf life if published as it is now. I don't think any of the science should be changed, and no new science is needed.

To increase the appeal (readability and accessibility) of the manuscript, I suggest considering the following major and minor comments.   

My major comment is on the handling of the data. Although the statistical analysis of cosmogenic nuclide is not my strength, I recently started to use the probabilistic cosmogenic age analysis tool (PCAAT) published by Dortch et al. (2022). I analyzed the 4 exposure ages from site 1 and it yielded the following two peaks: at 18.2 ± 1.5 ka (ext. & probability height is 0.26) and at 22.4 ± 1.8 ka (ext. & probability height is 0.11). Although the boulder JSD-21-04 is not a statistical outlier, can it be a geological outlier? This older exposure age from this boulder appears to be caused by inheritance/pre-exposure. In this case, how significant is the calculated mean age of 19.3 ± 2.1 ka in a geological/geomorphological context? In contrast, the single exposure age from site 2 seems to be “too young” with respect to regional context (cf. Figure 2). Can this boulder be exhumed? So, I suggest revisiting the calculation of the mean ages and expanding the regional significance of exposure ages (chapter 6.1). I propose to discuss the following topics in this subchapter: the lack of boulders (e.g., systematic removal of boulders during and after the agricultural revolution, for example, we know from the northern Swiss Alpine Foreland that farms were “cleared” [cf. Akçar et al., 2011]. I also know that historical documents on the companies which were hired to “clean the fields” in North America exist), the significance of geomorphic constraints, the interpretation of exposure ages (e.g., inheritance vs. exhumation) site by site, dating artifacts (e.g., PR calibration), etc. In brief, I think that the beautiful dataset presented in this manuscript deserves a more detailed discussion.

My next comment is on the discrepancy between cosmogenic nuclide chronology and 14C chronology. We also experienced a similar discrepancy in the dating of a historical rock avalanche in Mont Blanc Massif (Italy). The detailed analysis of the existing 14C chronology revealed that the 14C ages from bulk samples overestimated the timing of the RA, whereas the 14C from wood fragments were in agreement (Akçar et al., 2012; 2014). We went back and opened new trenches, there we compared the 14C ages from fossils and “bulk samples” (Hajdas et al., 2021). Based on our own experience, I recommend being careful when interpreting the 14C ages from bulk samples. Therefore, it might be appropriate to differentiate these in Figures 2 and 6 by using different symbology. The discrepancy could also be discussed in detail in the second part of the subchapter 6.1. In addition, the geomorphic context of each 14C site with respect to the ice sheet retreat is also important and needs to be addressed, especially in the discussions.

My last comment is on the regional retreat of the Laurentide Ice Sheet. I propose calculating a regional retreat rate based on cosmogenic 10Be chronology. To do that, mean or landform ages for each site (including the published chronologies such as in Figure 1) could be plotted over distance or latitude and a line could be regressed through the data points. I generally use the Bayesian regression code provided by Bender et al. (2016). In this way, a retreat rate with uncertainties could be calculated. Then, this rate can be compared to the modeled rates for Laurentide Ice Sheet and/or Antarctic/Greenlandic Ice Sheets today or in the past. These then could be appended to the discussions.

I congratulate the authors for this beautiful contribution and wish you good luck with the revisions,

With my best regards,

Minor comments:

Excessive use of “~”: Please clarify in the text what this symbol means and why it is used. I guess it indicates mean or landform ages.

Figure 1: Please extend the coverage by about 0.5 degrees to the south (down to 40.5 °N) to include the full LGM extent. Please indicate the full text of the states abbreviated on the map in the figure caption, especially for non-US readers. Why the symbol “~” is used? Does this show the landform/mean ages? Or it means “about”? In the case of the landform/mean ages, please indicate this abbreviation in the text and figure caption. In addition, how are these landform/mean ages calculated?

Figure 2: Could you please use a different format to differentiate 14C ages from bulk samples? Please see my major comment above for details.

Table 1: Please update the unit of the calibrated 14C ages. After calibration 14C ages are reported as years BP. Please add this to the table and correct the text accordingly.

Table 2: Please provide the size of the boulders in an additional column, at least the height.

Table 3: 10Be concentrations are reported in the order of 1e-4 and 1e-3. I guess in order of thousand and ten thousand atoms per gram. Please correct.

Figure 5: Please consider this density function plot by the PCAAT one. Please see my major comment above. Please indicate the mean ages of each site on the plot.

Figure 6: Please use a different symbol to differentiate organic 14C from “bulk” samples and fossils/organic fragments. Are all the plotted 14C ages recalibrated as the ones in Table 1? Please clarify and recalibrate if they are not already.

Figure 7 (New Figure): I suggest adding a new figure. Please consider re-analyzing the existing cosmogenic nuclide chronology with PCAAT and plotting the distance/Latitude vs “most probable” mean ages. Then, you can regress a line through the data and calculate a retreat rate. I generally use the Bayesian regression code provided by Bender et al. (2016).

Reviewer 2 Report

The deglaciation in North America has been a key phase of natural climate change. Robust data are needed to precisely reconstruct the melting of the ice sheet. This study adds valuable new data and discusses conflicting evidence derived from alternative methods. The figures are of high quality and help to better understand the current knowledge and implications of the new data.

The paper is well structured and written in a way so that a wide geoscience audience can understand it. Photographs illustrate the sample locations. The data are thoroughly tabulated, the methodogy clearly reported. Results and Discussion are separated, following the standard scientific approach.

After thorough review, I have no major suggestions for improvement. I would welcome to see this mansucript being swiftly published.

Author Response

Thank you for reviewing our article. We are glad that you were satisfied with the study as it stands and recognized the utility of this project.

Reviewer 3 Report

Dear Editor; Dear Authors

This is an interesting work, well written and well documented, both in terms of references to previous work and the method used (choice of samples, analytical technique).

But the purpose of this work being to determine the age of the samples (boulders, bedrock) using the analysis of 10Be to specify the chronology of the retreat of the ice sheet from central New England during the Last Glacial Maximum (LGM), I think the discussion should focus more on the atypical character of the sample 2. Only one sample (JSD-21-05 Boulder) characterizes the site 2, the estimated age of which is 12.5 ka. This value seems abnormal at this latitude (figure 6).

With in-depth critical work, this work could specify in what way the rapid retreat of the ice sheet occurred between latitudes 41 and 42°N. This discussion is all the more important as there is a disagreement between the ages given by 14C and Be10. As such, focusing the discussion on this observation would help the reader to better understand the causes and the mechanisms of the rapid retreat.

Minor recommendation:

Lines 146-147: At these latitudes, the climate was more influenced by the Gulf Stream than by the Atlantic Meridional Overturning Circulation (AMOC). Further explanation is desired.

Round 2

Reviewer 3 Report

The paper is interesting, the authors have provided some additional details. I think that it can be published as it is and that it will arouse the interest of the readers of Geosciences concerned with the subject addressed.